# Microenvironment regulation breaks the Faradaic efficiency-current density trade-off for electrocatalytic deuteration using $D_2O$

Meng He[1,4], Rui Li[1,4], Chuanqi Cheng[1,4], Cuibo Liu [1,2]✉ & Bin Zhang [1,3]✉

The high Faradaic efficiency (FE) of the electrocatalytic deuteration of organics with $D_2O$ at a large current density is significant for deuterated electrosynthesis. However, the FE and current density are the two ends of a seesaw because of the severe $D_2$ evolution side reaction at nearly industrial current densities. Herein, we report a combined scenario of a nanotip-enhanced electric field and surfactant-modified interface microenvironment to enable the electrocatalytic deuteration of arylacetonitrile in $D_2O$ with an 80% FE at $-100$ mA cm$^{-2}$. The increased concentration with low activation energy of arylacetonitrile due to the large electric field along the tips and the accelerated arylacetonitrile transfer and suppressed $D_2$ evolution by the surfactant-created deuterophobic microenvironment contribute to breaking the trade-off between a high FE and large current density. Furthermore, the application of our strategy in other deuteration reactions with improved Faradaic efficiencies at $-100$ mA cm$^{-2}$ rationalizes the design concept.

Owing to the kinetic isotope effect, deuterium (D) labelling has demonstrated important applications in the pharmaceutical industry, organic synthesis, and materials science[1–7]. Renewable electricity-powered deuteration using $D_2O$ provides an appealing D incorporation strategy due to its mild and decarbonization process, high reaction efficiency, and avoidance of utilizing other difficult-to-handle D sources (e.g., $D_2$, LiAlD$_4$, and CD$_3$OD)[8–19]. Typically, electrocatalytic deuteration reactions (EDRs) using $D_2O$ involve the in situ generation and utilization of active D atoms, and the $D_2$ evolution reaction (DER) is a major competitive reaction that lowers the Faradaic efficiencies (FEs) of target products. In particular, the restricted mass transfer and increased DER at high current densities make it highly challenging to achieve high FEs[10–18]. Therefore, designing an advanced electrocatalytic strategy to break the trade-off between high FEs and large current densities of EDRs using $D_2O$ is highly desirable but remains a great challenge (Fig. 1a).

Engineering electrocatalysts by doping, alloying, creating coordination-unsaturated sites, etc. is efficient at achieving high FEs, but the current densities are only a few to tens of mA cm$^{-2}$ (refs. 20–22). Recently, interface microenvironment modulation has proven to be effective at boosting electrocatalytic performance[23–30]. The addition of surfactants can accelerate mass transfer and create a hydrophobic interface microenvironment conducive to improving FEs[31–35]. In addition, the interface electric field also plays a significant role in electrocatalytic reactions. A high-curvature tip structure can enhance the local electric field[36,37], which further concentrates charged species or polar reactants to promote electrochemical transformations. Thus, we speculate that exploiting an optimized interfacial microenvironment by changing the interfacial hydrophilicity/hydrophobicity and local electric field will be a possible solution for achieving high FEs at high current densities. Furthermore, revealing the mechanism underlying the microenvironment modulation of EDRs using $D_2O$ is highly important but remains untouched.

Herein, high-curvature copper nanotips with unsaturated sites (Cu NTs) combined with butyl trimethyl ammonium bromide (BTAB)-modified electrode-electrolyte interface were designed (Fig. 1b). This comprehensive system can render the electrocatalytic

[1]Department of Chemistry, School of Science, Tianjin University, Tianjin 300072, China. [2]Institute of Molecular Plus, Tianjin University, Tianjin 300072, China. [3]Key Laboratory of Systems Bioengineering (Ministry of Education), Tianjin University, Tianjin 300072, China. [4]These authors contributed equally: Meng He, Rui Li, Chuanqi Cheng. ✉e-mail: cbliu@tju.edu.cn; bzhang@tju.edu.cn

**a  Process analysis of electrocatalytic deuteration of organics using D₂O and competitive D₂ evolution reaction**

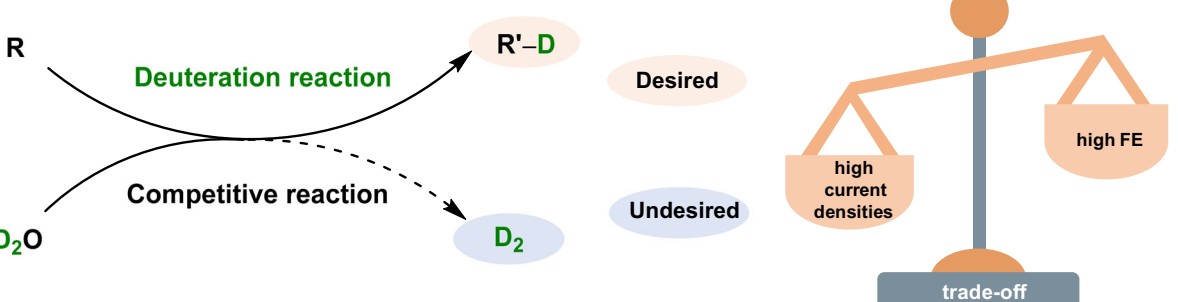

**b  Our synergistic strategies of tip-enhanced electric field and surfactant-tuned interface microenvironment**

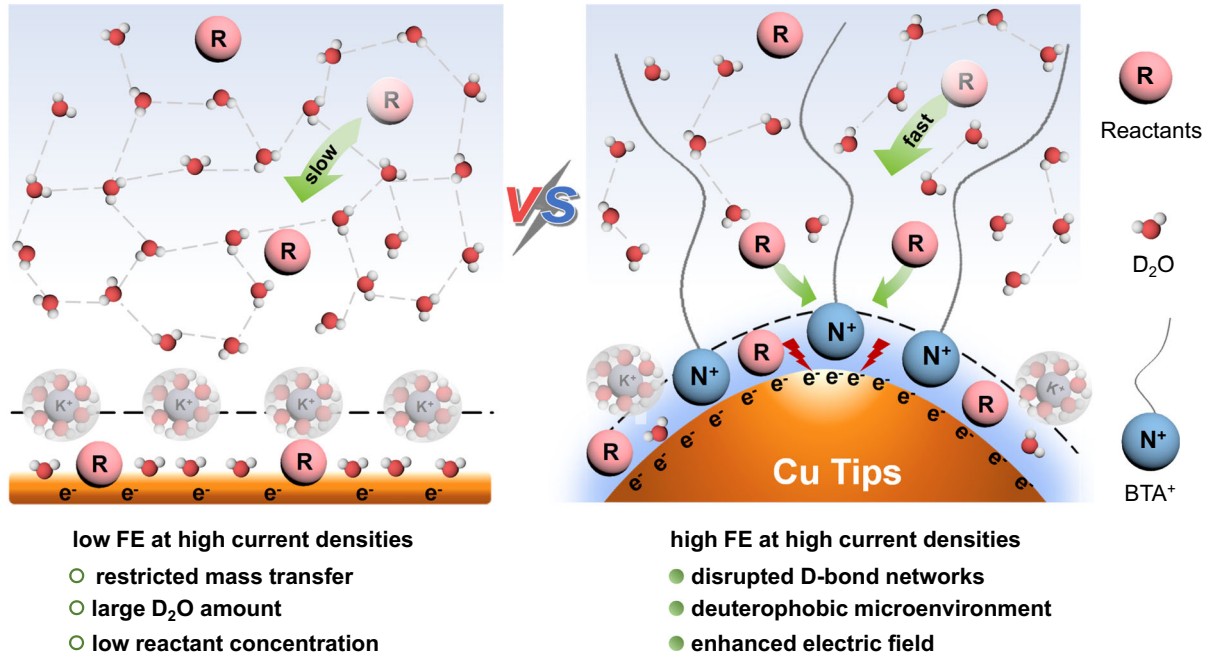

**Fig. 1 | Electrocatalytic deuteration of organics using D₂O. a** A low Faradaic efficiency (FE) at high current densities caused by severe D₂ evolution. **b** Our developed interfacial microenvironment control strategy involves changing the interfacial deuterophilic/deuterophobic property and local electric field to overcome the trade-off between a high FE and current densities.

---

deuteration of arylacetonitrile with an 80% FE at −100 mA cm$^{-2}$, greatly outperforming the corresponding Cu NTs without BTAB and Cu nanorods (NRs)/nanosheets (NSs) with BTAB. The theoretical and experimental results reveal that the adsorbed BTAB disrupts the hydrogen bond networks between water molecules and creates a deuterophobic microenvironment, promoting reactant transfer and suppressing DER. The nanotips display an enhanced local electronic field, small charge transfer resistance ($R_{ct}$), and low activation energy for deuteration, benefiting to the concentration of reactants near the electrode and accelerating the deuteration process. The wide substrate scope, gram-scale synthesis, and successful deuteration of other types of reactants, including alkynes, halides, *N*-heterocycles, and nitro compounds, with improved FEs at −100 mA cm$^{-2}$ highlight the universality of our strategy.

## Results
### Factors considered for microenvironment regulation
The electrocatalytic deuteration of arylacetonitriles to $\alpha,\beta$-deuterated arylethyl primary amines ($\alpha,\beta$-DAEPAs) in D₂O was selected as a model reaction because $\alpha,\beta$-DAEPAs serve as key precursors for synthesizing a variety of important deuterated drugs (Supplementary Fig. 1)[38–40]. To

achieve high FEs at high current densities, our consideration factors are listed below: (1) selecting materials with a large Gibbs free energy of hydrogen adsorption ($\Delta G_{H^*}$)[41,42] for suppressing the DER, (2) improving the mass transfer to accelerate the deuteration kinetics, and (3) concentrating reactants around the electrode surface to increase the reaction rates. Cu-based nanomaterials have been intensively studied as ideal candidates for many electrocatalytic transformations[43–46], and high FEs have been obtained for the hydrogenation of NO₃⁻, CO₂, and acetylene at high current densities over Cu electrocatalysts[47–50]. Thus, we selected nano-Cu materials as electrocatalysts for the electrocatalytic deuteration of arylacetonitriles. In addition, the mass transfer limitation may be ascribed to the formation of deuterium bond networks between D₂O molecules that restrict reactant migration to the electrode surface. Quaternary ammonium (QA) salts and organic sulfur compounds are reported to be very useful in promoting mass transfer in electrocatalytic transformations involving water[32,51–53]; thus, the reaction activity and/or product selectivity are improved. The results of the ab initio molecular dynamics (AIMD) simulation (Supplementary Data 1 for the optimized DFT computational models) further demonstrate that the number of hydrogen bonds is significantly reduced in the presence of BTAB (Fig. 2a and Supplementary Fig. 2 and Note 1),

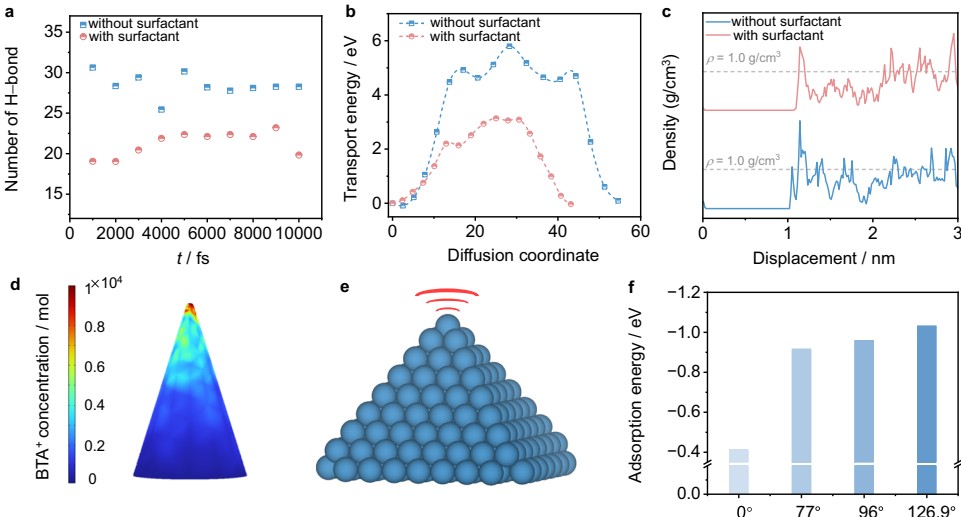

**Fig. 2 | Theoretical results guiding the design of integrating QA surfactant with nanotips for the electrocatalytic deuteration of arylacetonitriles using $D_2O$.** Comparison of **a** the number of hydrogen bonds and **b** the transport energy of **1a** in 0.5 M $K_2CO_3$ with or without the BTAB surfactant. **c** Mass density of $H_2O$ molecules versus distance above the electrode surface. **d** Surface BTA⁺ density distributions on the surface of the Cu NTs. **e** Schematic illustration of the Cu NTs. **f** Adsorption energies of **1a** for different crystal surface pinch angles.

suggesting the breakdown of the hydrogen bond networks. As a result, the transport energy barrier of *p*-methoxyphenylacetonitrile (**1a**, a model substrate) is markedly reduced, which is advantageous to the migration of **1a** to the electrode surface, resulting in an increase in their concentration and enhancing the activity of electrocatalytic deuteration (Fig. 2b and Supplementary Fig. 3 and Note 2). In addition, the enlarged distance between the $H_2O$ molecules and the electrode suggests a reduced amount of $H_2O$ molecules in the Helmholtz layer (Fig. 2c), which indicates that the introduction of BTAB can also reduce the coverage of interface $H_2O$ to further prohibit the $H_2$ evolution reaction.

Moreover, enhancing the local electric field to strengthen the interaction between the electrode and reactant is favourable for increasing the concentration of the reactant on the electrode surface, thereby accelerating the electrocatalytic transformations[54–56]. The results of the finite element method simulation show that the electric field intensity along the tips is greatly enhanced when the cone is sharpened from 50 (left) to 5 nm (right) (Supplementary Fig. 4a and Note 3). This difference is attributed to electrostatic repulsion, which causes free electrons to migrate to the tips of the Cu electrode[36]. The Gouy–Chapman–Stern model was used to estimate the effect of locally enhanced electric fields on the concentration of the adsorbed butyl trimethyl ammonium cation (BTA⁺). The mapped concentration of surface-adsorbed BTA⁺ in the Helmholtz layer adjacent to the Cu electrode surface increases gradually with decreasing curvature radius (Fig. 2d), which is favourable for increasing the **1a** concentration and lowering the $D_2O$ amount near the electrode, thereby promoting the deuteration of **1a** and inhibiting the DER. In addition, the Cu electrode surface has abundant crystalline prisms as the radius of curvature of the tip decreases (Fig. 2e and Supplementary Fig. 4b). It is easier to absorb **1a** at the crystalline prisms than at the flat structure, and the adsorption energy of **1a** increases with increasing crystal surface pinch angle (Fig. 2f), suggesting that nanotips can easily absorb **1a** and are conducive to improving **1a** deuteration.

Based on the above considerations and theoretical results, the design of high-curvature Cu nanotips coupled with a QA surfactant-modified electrode-electrolyte interface would be a good solution for addressing the trade-off problem between high FEs and high current densities of EDRs using $D_2O$.

## Electrocatalytic deuteration of 1a using $D_2O$ over Cu nanotips modified with butyl trimethyl ammonium bromide as a surfactant

Self-supported Cu NTs with abundant coordination-unsaturated sites were synthesized by the electrochemical reduction of CuO NTs at −1.0 V vs. Hg/HgO (all the potentials in this work refer to Hg/HgO unless otherwise stated and were calibrated with a reversible hydrogen electrode) (Supplementary Figs. 5, 6a and Notes 4, 5). Scanning electron microscopy (SEM) and transmission electron microscopy (TEM) images reveal the nanotip structure of the obtained Cu NTs (Fig. 3a and Supplementary Fig. 6b). In situ Raman spectra demonstrates the electrochemical reduction process of CuO→$Cu_2O$→Cu, and the X-ray photoelectron spectroscopy (XPS) and x-ray diffraction (XRD) results validate the formation of metallic Cu after the electroreduction of CuO (Supplementary Fig. 6c–e)[21]. Furthermore, X-ray absorption near-edge structure (XANES) and extended X-ray absorption fine structure (EXAFS) characterization confirm the presence of coordination-unsaturated sites in Cu NTs with a Cu−Cu coordination number of 7.3, which is smaller than that of Cu foil (12) (Fig. 3b and Supplementary Fig. 7, Note 6, and Supplementary Table 1). These characterization results illustrate that coordination-unsaturated sites easily form in high-curvature nanotip materials.

Typically, the deuteration reaction is carried out in a divided H-type reactor by adding **1a** to a mixed solution of dioxane (Diox) and 0.5 M $K_2CO_3$ in $D_2O$ (2:5 v/v, 7 mL) over a Cu NT cathode. Before starting the reaction, we adopted cyclic voltammetry (CV) to study the deuteration process. The CV curve (red line) reveals an obvious reduction peak centered at approximately −0.78 V after the addition of **1a** (Fig. 3c), which is more positive than the reduction potential of $D_2O$ at approximately −0.98 V (blue line). This finding implies that **1a** is easier to be electroreduced than $D_2O$. Potential screening experiments show that the optimal results, including a 91% yield and 90% FE of α,β-deuterated amine product **2a**, can be obtained at −1.3 V (Supplementary Figs. 8, 9). Additionally, both the yield and FE of **2a** are higher over Cu NTs than those over Cu-$H_2$ NTs, which have fewer coordination-unsaturated sites at the same potentials (Fig. 3b, Supplementary Fig. 9, and Note 7). These results demonstrate the promoting effect of more low-coordination sites on the deuteration of **1a**. However, Fig. 3d shows that the FE of **2a** (blank) is only 60% over the Cu NTs at −100 mA cm⁻² under the theoretical Coulomb charge. The

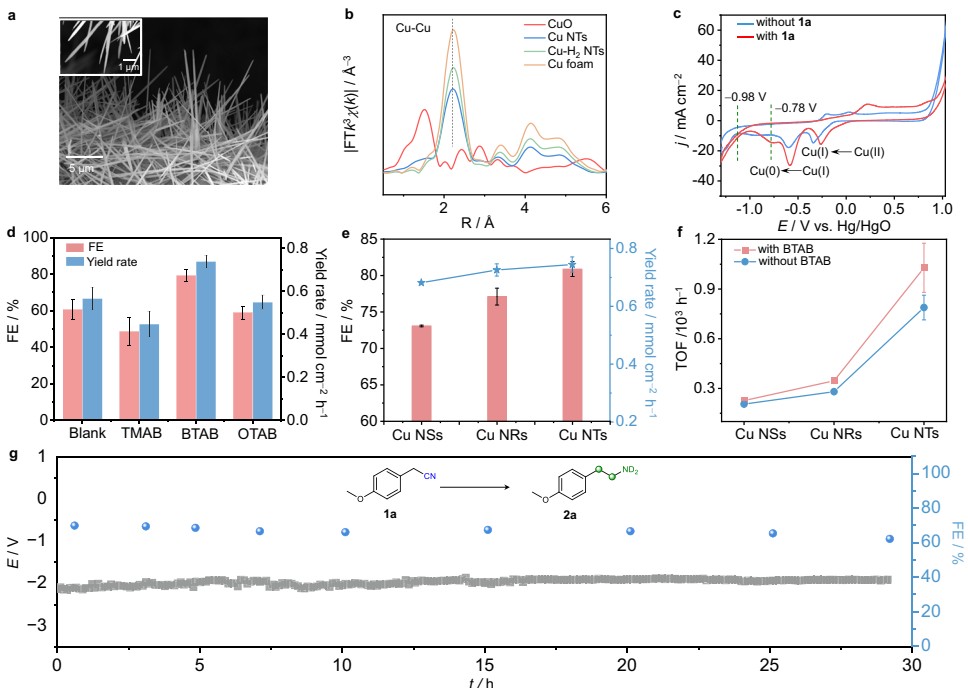

**Fig. 3 | Characterization of the Cu NTs and performance of the electrocatalytic deuteration of 1a with D₂O. a** SEM image of the Cu NTs. **b** EXAFS spectra of the Cu NTs, Cu-H₂ NTs and Cu foam. **c** CV curves of Cu NTs at a scan rate of 3 mV s⁻¹ in Diox/ 0.5 M K₂CO₃ (2:5 v/v, 7 mL) with and without **1a**. **d** Comparison of the FEs and yield rates of **2a** obtained by adding different surfactants at −100 mA cm⁻². **e** FEs and yield rates of **2a** at −100 mA cm⁻² over Cu NTs, NRs, and NSs with BTAB. **f** TOF values of **2a** at −100 mA cm⁻² with or without BTAB over Cu NTs, NRs, and NSs. **g** Durability tests of the Cu NTs with BTAB in a flow cell (R: 1.4 ± 0.23 Ω). The error bars correspond to the standard deviation of at least three independent measurements, and the centre value for the error bars is the average of the three independent measurements.

reduced **2a** FE may be mainly ascribed to the restricted mass transfer of **1a** to the electrode surface at high current densities, which slows down the deuteration of **1a**, thus leading to an increase in the DER by the large amount of D₂O in the Helmholtz layer. To improve the FEs, QA surfactants with the same head groups but different tail chains were added to the reaction system under other identified conditions according to our considerations and theoretical results (Supplementary Fig. 10). As expected, the introduction of BTAB, which has a medium-length tail, can increase the FE up to 80% and requires the lowest cell voltage at −100 mA cm⁻² (Fig. 3d and Supplementary Fig. 11a). The calculated reaction rate approaches 0.74 mmol h⁻¹ cm⁻², which is 38.9 times faster than the reaction rate (0.019 mmol h⁻¹ cm⁻²) of our previous work[14] (Fig. 3d). In contrast, the use of tetramethyl ammonium bromide (TMAB) and octadecyl trimethyl ammonium bromide (OTAB) had no obvious or even negative effect on the deuteration of **1a**. We speculate that the smallest TMAB may block more active sites on Cu and be ineffective at breaking hydrogen bond networks, whereas the longer alkyl chain of OTAB may affect the migration of **1a**, thus reducing the reaction efficiency. A detailed screening of the concentrations of the QA surfactant reveal that 1.0 mM BTAB gives the best result for **1a** deuteration (Supplementary Fig. 11b). The above results demonstrate that a suitable amount of QA surfactant with an appropriate chain length tail can improve the FEs of electrocatalytic deuteration at high current densities. Furthermore, Cu NRs and Cu NSs were also synthesized to investigate the morphological effect of Cu electrocatalysts on the electrocatalytic deuteration of **1a** (Supplementary Fig. 12 and Note 8). The highest FE and yield of **2a** over the Cu NTs imply intrinsically high activity (Fig. 3e). Moreover, compared with those of the Cu NRs and NSs, the turnover frequency (TOF) of **2a** exhibits the most obvious increase compared with that of the Cu NTs after the addition of BTAB (Fig. 3f and Supplementary Fig. 13 and Note 9), which may be ascribed to the larger local electric field of the Cu NTs that can concentrate more BTAB, thus enhancing **1a**

deuteration. These results confirm that the high-curvature nanotip structure accelerates the electrocatalytic deuteration of **1a**, consistent with our theoretical prediction. Furthermore, -30 h of continuous electrolysis of **1a** over Cu NTs was conducted by adopting a flow reactor at −100 mA cm⁻² (Fig. 3g and Supplementary Fig. 14 and Note 10). The relatively steady cell potential, only a slight decrease in the FE (<7%), and no appreciable alteration in the morphology and XRD pattern of the used sample reveal the robust durability of the Cu NTs (Supplementary Fig. 15 and Note 11). Consequently, the Cu NTs show excellent activity and stability toward the electrocatalytic deuteration of **1a** using D₂O.

## Unveiling the roles of butyl trimethyl ammonium bromide and nanotips

Several in situ and ex situ spectroscopy characterizations, electrochemical tests, and control experiments were conducted to explore the combined promotional effects of the QA surfactant and nanotips on the electrocatalytic deuteration of **1a** using D₂O.

**Exploring the roles of BTAB.** First, in situ attenuated total reflection Fourier transform infrared (FTIR) spectroscopy reveals an increase in the intensity and blueshift of the C−H stretching mode ($\nu_{C-H}$) of the BTAB head group (N−CH₃) with the application of additional negative potentials (Supplementary Figs. 16a, b and Supplementary Note 12). Additionally, the N 1s XPS spectrum displays an evident peak located at 399.1 eV, which is attributed to the N−Cu bond[57], after the Cu NTs were treated with BTAB (Supplementary Fig. 16c). These results indicate that BTAB adsorbs on the Cu NTs surface under electrochemical conditions. Second, the adsorption of D₂O molecules in the absence and presence of BTAB was also investigated. The peaks centered at 2525 and 1245 cm⁻¹ are assigned to the O−D stretching mode ($\nu_{O-D}$) and the D−OD bending mode ($\delta_{D-OD}$) of D₂O[58], respectively (Fig. 4a). The intensity of these peaks gradually increases as the applied potential

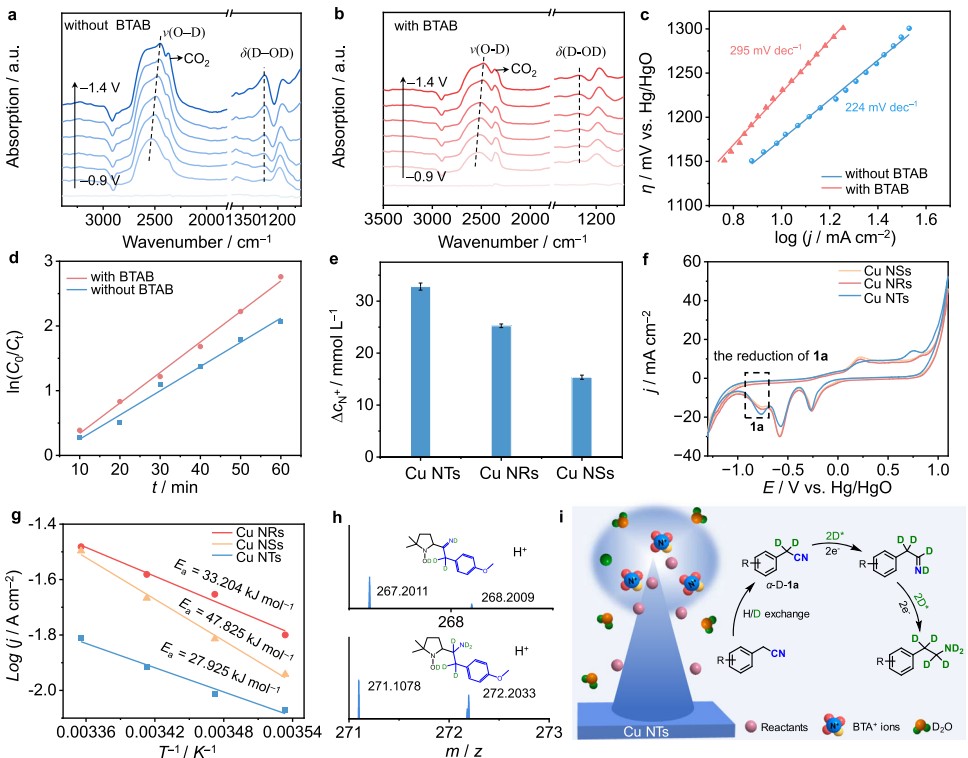

**Fig. 4 | Mechanistic studies.** In situ ATR-FTIR spectra under various bias potentials for **a** pure $K_2CO_3$ and **b** the BTAB-containing system without **1a** over Cu NTs. **c** Tafel slopes of the DER over Cu NTs in 0.5 M $K_2CO_3$ with and without BTAB. **d** Kinetic investigations of Cu NTs for the deuteration of **1a** with and without BTAB. **e** Field-induced BTA$^+$ concentration loss in the electrolytes caused by the use of Cu NTs, Cu NRs, and Cu NSs. Error bars correspond to the standard deviation of at least three independent measurements, and the centre value for the error bars is the average of the three independent measurements. **f** CV curves of the Cu NTs, NRs, and NSs at a scan rate of 3 mV s$^{-1}$ in Diox/0.5 M $K_2CO_3$ with **1a**. **g** Arrhenius fitting plots of Cu NTs, Cu NRs, and Cu NSs for the dependence of the reaction rate for the deuteration of **1a** on temperature. **h** HR-MS results of the possible intermediates during electrocatalytic deuteration of **1a** with $D_2O$ over Cu NTs by adding the trapping agent DMPO. **i** A proposed possible mechanism.

becomes more negative, suggesting the adsorption of $D_2O$ on Cu. However, the intensity of these two peaks becomes weaker after the addition of BTAB and increases slowly with potential (Fig. 4b). This may suggest that the BTAB adsorbed on the Cu NTs surface can repel the $D_2O$ molecules in the Helmholtz layer, thereby reducing the amount of $D_2O$. Moreover, we observe a slight redshift of the $v_{O-D}$ mode (2478 vs. 2445 cm$^{-1}$) in the presence of BTAB (Supplementary Figs. 17a, b and Note 13), suggesting weaker adsorption of $D_2O$ on the Cu NTs. This difference hints at the large distance between the D atom and the electrode surface because of the orientation changes in the interfacial $D_2O$ molecules[53], in agreement with the DFT calculations (Fig. 2c). In addition, the larger contact angle of 122.88° for the Cu NTs electrode with BTAB indicates the hydrophobic role of BTAB (Supplementary Fig. 18). Third, the Nyquist plots and Bode plots obtained by electrochemical impedance spectroscopy (EIS) measurements display a larger semicircle and phase angle at the same applied potential for the DER when BTAB is added (Supplementary Figs. 19, 20 and Note 14). The calculated charge transfer resistance ($R_{ct}$) according to the equivalent circuit model (Supplementary Table 2) is larger for the DER, indicating slower electron transfer for the BTAB-modified interface. Therefore, the Tafel slope for the DER increases from 224 to 295 mV dec$^{-1}$, and a more negative potential is needed to achieve a benchmark current density of $-10$ mA cm$^{-2}$ after adding BTAB (Fig. 4c and Supplementary Fig. 21 and Note 15), implying a suppressed DER. Fourth, the outer Helmholtz layer changes when BTA$^+$ adsorbs on the electrode surface under electrochemical conditions, which leads to a shortening of the Helmholtz layer (normally ~3.0 Å) due to the smaller radius of the BTA$^+$ head group (2.79 Å)[56]. In addition, our theoretical result in Fig. 2a reveals the breakdown of the hydrogen bond networks

by the introduction of the QA surfactant. These positive factors can accelerate the migration of reactants toward the electrode surface through dipole interactions between the reactant and the electrode, thus increasing their local concentration. As shown in Fig. 4d, the electrochemical kinetic data is consistent with the pseudo-first-order kinetics model. This shows that the reaction rate is related to the concentration of **1a** and that the introduction of BTAB can effectively improve the kinetics of the deuteration of **1a**. This explanation is further validated by the obvious increase in the peak at **1a** in the CV curves after introducing BTAB (Supplementary Fig. 22 and Note 16). A concentrated **1a** near the Cu surface is favourable for electron transfer from Cu to **1a** to further promote its deuteration, which is reflected by the smaller $R_{ct}$ value and phase angle (Supplementary Fig. 23). The above results suggest that the presence of BTAB can disrupt deuterium bond networks, create a deuterophobic environment, and increase the **1a** concentration in the Helmholtz layer, thereby improving the deuteration of **1a** and suppressing the DER.

**Validation of the roles of the nanotip structure.** Inductively coupled plasma atomic emission spectroscopy (ICP–AES) was used to measure the amount of BTA$^+$ adsorbed away from the electrolytes by the Cu NTs, NRs, and NSs under the same potential. Figure 4e shows that the high-curvature Cu NTs have the largest electric-field-induced locally absorbed BTA$^+$ concentration, although Cu NTs have the least active sites (Supplementary Fig. 13). This result, coupled with the finite element simulation of the tip-enhanced electric field intensity (Supplementary Fig. 4a), supports the local occurrence of a large electric-field-induced reagent concentration effect by the Cu NTs, which agrees well with our theoretical results (Fig. 2d). As a consequence, more **1a** is

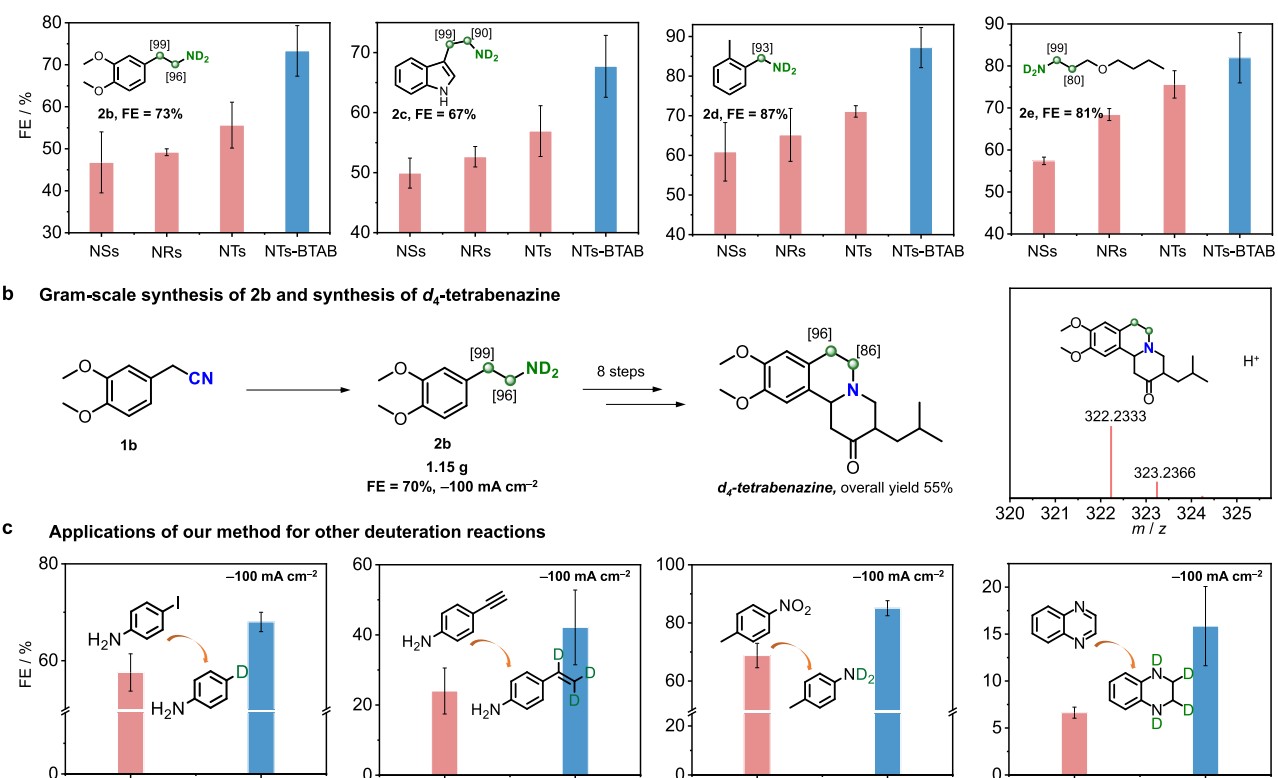

**Fig. 5 | Methodology universality. a** Electrocatalytic deuteration of other aryla-cetonitriles using $D_2O$ over Cu NSs, NRs, and NTs without and with BTAB (the data in square brackets represent the deuterium ratios). **b** Gram-scale synthesis of **2b** at a current density of −100 mA cm$^{-2}$ and the synthesis of $d_4$-tetrabenazine. **c** Other types of electrocatalytic deuteration using $D_2O$ over Cu NTs with and without BTAB. The error bars correspond to the standard deviation of at least three independent measurements, and the centre value for the error bars is the average of the three independent measurements.

concentrated over the Cu NTs than the Cu NRs and NSs, as demonstrated by the larger reduction peak of **1a** over the Cu NTs in the CV curves (Fig. 4f). Therefore, an increased concentration of **1a** can better match the deuteration step at a high current density, accelerating **1a** deuteration while inhibiting the DER. In addition, compared with those of Cu NRs and NSs, the $R_{ct}$ value of Cu NTs for **1a** deuteration is the lowest, suggesting the faster electron transfer from Cu NTs to **1a** (Supplementary Fig. 24). Furthermore, the electrochemical activation energies were obtained from the slope of the Arrhenius plots at −1.25 V. The activation energy for Cu NTs is lower than that for Cu NRs and NSs, highlighting the more favourable thermodynamics of **1a** deuteration over Cu NTs (Fig. 4g and Supplementary Fig. 25). Therefore, the kinetically improved mass and charge transfer and the thermodynamically favourable **1a** activation of Cu NTs by the large electric field produced account for the enhancement in the deuteration of **1a** with high FEs at high current densities.

**Identifying the key intermediates.** After clarifying the roles of the QA surfactant and nanotips in promoting the electrocatalytic deuteration of **1a** using $D_2O$, we further investigated the adsorption of **1a** and the key intermediates during the deuteration reaction. In situ Raman spectroscopy reveals a visible redshift of the C≡N vibration ($\nu_{C≡N}$) of **1a** in the presence of the Cu NTs compared with that of pure **1a** (Supplementary Fig. 26), implying that the C≡N group adsorbed on the Cu surface, as demonstrated by the Tian group[59]. The intensity of $\nu_{C≡N}$ first increases, which may be ascribed to the field effect-enhanced concentration of **1a**, and then decreases because of the **1a** conversion as electrolysis proceeds. Usually, the hydrogenation of nitriles proceeds through an imine intermediate[14,45,46]. However, no signal related to deuterated imine is detected, which may be due to the easier

deuteration of imines than of **1a**, as supported by the DFT calculations (Supplementary Fig. 27 and Note 17). Furthermore, electron paramagnetic resonance (EPR) measurements obtained by adding a 5,5-dimethyl-1-pyrroline-*N*-oxide (DMPO) trapping agent to the reaction system display characteristic hyperfine structural signals corresponding to the DMPO-D and DMPO-C spin adducts (Supplementary Fig. 28a and Note 18). The mass-to-charge ratios ($m/z$) of DMPO-D and DMPO-C were further confirmed by high-resolution mass spectrometry (HR-MS) (Fig. 4h and Supplementary Fig. 28b). These results demonstrate the formation of deuterium and carbon radicals during **1a** deuteration using $D_2O$.

Based on the above results, a possible mechanism for the one-pot deuteration of **1a** to **2a** using $D_2O$ is proposed (Fig. 4i). The rapid $K_2CO_3$-assisted α-C−H to α-C−D exchange of **1a** with $D_2O$ generates α-deuterated **1a** (denoted as α-D-**1a**) (Supplementary Fig. 29). When electrolysis begins, BTA$^+$ is rapidly adsorbed to the Cu NT surface due to the enhanced electric fields along the tips, which can accelerate the migration of α-D-**1a** to the electrode surface and repel $D_2O$ from the Helmholtz layer. Then, the electrocatalytic α-D-**1a** proceeds via a stepwise D radical-involved process to form the deuterated amine **2a**. Finally, **2a** desorbs from the surface of the Cu NTs with the regeneration of the catalytic sites for the next reaction cycles.

## Methodology universality

Our strategy is applicable to the electrocatalytic deuteration of a series of arylacetonitriles, aryl nitriles, and aliphatic nitriles by using $D_2O$ at −100 mA cm$^{-2}$ (**2b**-**2s**; Fig. 5a and Supplementary Table 3), suggesting the generality of our method. The higher FEs of the deuterated amine products obtained over Cu NTs in the presence of BTAB imply the synergistic promoting effect of the surface QA surfactant and nanotips

(Fig. 5a). A comparison of the synthesis of α,β-deuterated arylethyl primary amines between the reported methods and our method is provided in Supplementary Fig. 31. Moreover, the scale-up synthesis of **2b** (1.15 g) by adopting a flow reactor at −100 mA cm⁻² with a 70% FE can be accomplished (Fig. 5b), demonstrating the potential utility of our method. By using **2b** as the starting material, D-incorporated $d_4$-tetrabenazine, which has a 55% overall yield, was synthesized for the first time according to similar reported procedures[60,61]. D installed at the *N*-heterocycle of tetrabenazine may provide an alternative to further enhance the activity and metabolic stability of tetrabenazine for the treatment of chorea associated with Huntington's disease, which is expected to complement deutetrabenazine (Austedo™) (Supplementary Fig. 30 and Notes 19, 20) that bears two −OCD₃ moieties[62]. Furthermore, our method is also effective for the electrocatalytic deuteration of the C−I, C≡C, −NO₂, and *N*-heterocycle moieties to obtain improved FEs at −100 mA cm⁻². These encouraging results further demonstrate the general applicability of the synergistic effect of the QA surfactant and nanotips in promoting deuterated synthesis using D₂O (Fig. 5c and Supplementary Figs. 31, 32 and Note 21).

## Discussion

In summary, electrocatalytic deuteration by using D₂O as the D source provides a promising strategy for the synthesis of deuterated molecules; however, this method suffers from low FEs at high current densities, causing more energy consumption and low reaction rates. We demonstrate a combined microenvironment regulation strategy of high-curvature Cu nanotips with a BTAB-modified interface that strongly promotes the electrocatalytic deuteration of arylacetonitriles with D₂O, resulting in an FE of 80% at −100 mA cm⁻². In situ ATR-FTIR spectra showed that the addition of BTAB results in a deuterophobic interfacial microenvironment that reduces the amount of D₂O and facilitates the transfer of nitrile reactants. ICP–AES and CV tests confirmed that the nanotips can easily concentrate the cations of BTAB, which further accelerates the migration of nitriles to the Helmholtz layer. In addition, electrochemical impedance and activation energy tests revealed the low charge transfer resistance and low activation energy of nitrile deuteration over Cu NTs, thus promoting the deuteration of nitriles. Furthermore, deuterium and carbon radical intermediates were confirmed by EPR and HR-MS, and a deuterium radical-involved stepwise deuteration process was proposed. Our method was also efficient at deuterating a wide range of nitriles and other reducible groups to afford high FEs at −100 mA cm⁻², demonstrating the universality of the methodology. Moreover, the combined system could improve the FEs and yield rate of the electrocatalytic hydrogenation of **1a** using H₂O for hydrogenation (Supplementary Fig. 33 and Note 22). However, the lower zero point energy makes it harder to cleave the D−OD bond. Therefore, deuteration reactions start at a more negative potential than hydrogenation reactions (Supplementary Fig. 34 and Note 23)[63]. Our work not only provides a combined interface microenvironment regulation strategy to solve the challenge of the high FEs of multiple deuteration reactions at nearly industrial current densities by the synergy of surfactants and local electric fields but also offers a paradigm for other water-involved electrocatalytic transformations, such as electrocatalytic hydrogenation of NO₃⁻ and CO₂ and C−N coupling reactions to achieve high FEs at high current densities.

## Methods

### Materials

All chemicals used in the experiments were analytically pure and used without further purification. NaOH (Reagent grade, 97%), (NH₄)₂S₂O₈ (Puriss, 99%), K₂CO₃ (Analytic reagent, 99%), arylacetonitrile (ACS regent, 99.5%), 4-iodoaniline (Reagent grade, 98%), 4-ethynylaniline (HPLC, 99.4%), *p*-Nitrotoluene (Analytic reagent, 99%), quinoxaline

(Puriss, 99%), D₂O (HPLC, 99.9%), and 1,4-dioxane (HPLC, 99.8%) were purchased from Aladdin. A Nafion 117 exchange membrane was purchased from DuPont. A Hg/HgO (3 M) reference electrode (diameter 3.8 mm) was purchased from Shanghai Chuxi Industrial Co., Ltd. Cu foam (thickness 1.0 mm) was purchased from Guangzhou Guangjiayuan Industrial Co., Ltd.

### Synthesis of CuO nanotips (NTs)

Self-supported CuO NTs were synthesized by a wet chemical process[64]. First, commercial Cu foam (3 cm × 1 cm × 0.1 cm) was ultrasonically treated with acetone, 3.0 M HCl solution, and deionized water (DIW) for 15−20 min. Then, the treated Cu foam was immersed in an aqueous solution of 2.0 M sodium hydroxide and 0.15 M ammonium persulfate for 7 min without stirring. Then, the Cu foam covered with the blue Cu(OH)₂ NTs was removed from the solution, washed with DIW and absolute ethanol, and dried at room temperature. Finally, the Cu(OH)₂ NTs/Cu foam was put into a porcelain boat, heated at 150 °C for 2 h at a heating rate of 1 °C min⁻¹, and cooled to room temperature to obtain the CuO NTs.

### In situ electroreduction of CuO NTs to coordination-unsaturated Cu NTs

The Cu NTs were synthesized in situ via electroreduction of CuO NTs in a divided three-electrode system using 0.5 M K₂CO₃ solution as the electrolyte at −1.0 V vs. Hg/HgO for 30 min to ensure the complete disappearance of the reduction peak. The as-prepared Cu NTs with an exposed surface area of 1.0 cm² served as the working electrode for implementing the electrocatalytic deuteration reactions using D₂O.

### Electrochemical measurements

Linear sweep voltammetry (LSV) and chronoamperometry were performed using an electrochemical workstation (CS150H). The electrochemical deuterium measurements were carried out in a divided three-electrode electrochemical cell. The as-prepared electrocatalyst, carbon rod, and Hg/HgO were used as the working electrode, counter electrode, and reference electrode, respectively. (Note that the Hg/HgO electrode was calibrated with respect to a reversible hydrogen electrode in a high-purity hydrogen-saturated electrolyte with a Pt foil as the working electrode.) The cathodic and anodic chambers were separated by a Nafion 117 membrane. A 0.5 M D₂O solution of K₂CO₃ (5.0 mL) was added to the cathodic and anodic cells. Electrolyte solutions were prepared when used. Then, a certain amount of surfactant solution and 0.4 mmol of reactants dissolved in 2.0 mL of 1,4-dioxane were added to the cathodic cell under stirring (600 rpm) to form a homogeneous solution, and chronoamperometry was carried out at −100 mA cm⁻² under the theoretically required Coulomb charge for full conversion of 0.4 mmol of reactants. The addition of the right amount of 1,4-dioxane to the electrolyte improved the solubility of the organic substrates, which is beneficial for the deuteration reactions. All the potentials in this work were referred to as Hg/HgO without *iR* correction unless otherwise stated, and the *R* values were 1.4 ± 0.23 Ω. After the reaction was complete, 0.05 mmol of dodecane was added to the reaction mixture as an internal standard. Then, the solution at the cathodic cell was extracted with dichloromethane (DCM) and dried with anhydrous Na₂SO₄. The DCM was removed, and the products were tested by GC to calculate the yield of the amine products. In addition, the organic phase was treated with 3.0 M cyclopentyl methyl ether hydrochloric acid solution, and the precipitated solid product was filtered to calculate the isolated yields of the products, which were further subjected to NMR spectroscopy. The yield and FE, yield rate and TOF (**2a** as an example) were determined for the whole process of deuteration using dodecane as an internal standard and calculated using Eq. (1)−(4) below. The deuterated ratios of the

products were determined by $^1$H NMR according to Eq. (5)−(6).

$$\text{Yield}(\%) = \frac{n(\text{obtained products})}{n(\text{theoretically formed products})} \times 100\% \quad (1)$$

$$\text{FE}(\%) = \frac{4 \times n \times 96485}{Q(\text{theoretically values})} \times 100\% \quad (2)$$

$$\text{Yield rate}(\%) = \frac{n(\text{obtained products})}{t(\text{time required for the reaction})} \times 100\% \quad (3)$$

$$\text{TOF}(\text{s}^{-1}) = \frac{\text{The amount of produced amine molecules per site}}{\text{the amount of Cu sites}} \quad (4)$$

$$\text{Deuterium incorporation at } \alpha-\text{position}(\%) = 100\% - \frac{\text{area}(\text{R}-\text{CH}_2-\text{CD}_2-\text{NH}_2\cdot\text{HCl})}{2} \times 100\% \quad (5)$$

$$\text{Deuterium incorporation at } \beta-\text{position}(\%) = 100\% - \frac{\text{area}(\text{R}-\text{CD}_2-\text{CH}_2-\text{NH}_2\cdot\text{HCl})}{2} \times 100\% \quad (6)$$

## Characterizations

SEM images were taken with an FEI Apreo S LoVac SEM (1 kV). TEM images were obtained with an FEI Tecnai G2 F20 microscope. The X-ray diffraction (XRD) patterns of the products were analysed in the range of 10° to 90° at a scan rate of 20° min$^{-1}$ using a Rigaku Smartlab9KW diffraction system with a Cu $K\alpha$ source ($\lambda = 1.54056$ Å). X-ray photoelectron spectroscopy (XPS) was performed on a Thermo Fisher ESCALAB-250Xi photoelectron spectrometer using a monochromatic Al $K\alpha$ X-ray beam (1486.60 eV). All the peaks were calibrated with the C 1 s spectrum at a binding energy of 284.8 eV. The X-ray absorption spectra (XAS), including XANES and EXAFS of the Cu $K$-edge, were obtained under an ultrahigh vacuum at the 1W1B beamline of the Beijing Synchrotron Radiation Facility (BSRF). The XAS spectra were analysed with the ATHENA software package. The NMR spectra were recorded on a JEOL JNM-ECZ400S/L1 instrument at 400 MHz ($^1$H NMR) and 101 MHz ($^{13}$C NMR) with DMSO-$d_6$ and CDCl$_3$ as the solvents. Chemical shifts were reported in parts per million (ppm) downfield from the internal tetramethylsilane. The multiplicity is indicated as follows: s (singlet), d (doublet), t (triplet), m (multiplet), and br (broad). The coupling constants are reported in hertz (Hz). The quantitative analysis of the liquid products was conducted by a gas chromatograph (GC, Agilent 7890 A) with thermal conductivity (TCD), a flame ionization detector (FID), and an HP-5MS capillary column (0.25 mm in diameter, 30 m in length). Identification of the reactants and products was performed using gas chromatography–mass spectrometry (Agilent, 8860GC-5977MS) with an HP-5MS capillary column (0.25 mm in diameter, 30 m in length). The injection temperature was set at 300 °C. Nitrogen was used as the carrier gas at 1.5 mL min$^{-1}$. Accurate mass measurements of the products were obtained via high-resolution mass spectrometry (HR-MS, ESI, positive mode) on an Agilent 6550 QTOF. Hydrogen and carbon radicals were investigated via electron paramagnetic resonance (EPR) spectroscopy (JES-FA200, JEOL, Japan).

## In situ Raman spectroscopy

Raman spectroscopy was performed on the aforementioned Renishaw inVia reflex Raman microscope under excitation at 532 nm (for the Cu catalyst) or 633 nm (for the organic reactant) by using an in situ electrochemistry method. The electrolytic cell was constructed in-house by Teflon with a piece of round quartz glass as the cover to protect the objective. The working electrode was set to maintain the plane of the sample perpendicular to the incident laser. Pt wire was used as the counter electrode, and Hg/HgO was used as the reference electrode. A mixed solution of Diox/0.5 M K$_2$CO$_3$ with 0.4 mmol of **1a** was added to the cathodic cell. The Raman spectra were recorded by using chronoamperometry at −1.3 V.

## Electrochemical in situ ATR-FTIR tests

Electrochemical in situ ATR-FTIR measurements were performed on a Linglu Instruments ECIR-II cell mounted on a Pike Veemax III ATR with a single bounce silicon crystal covered with an Au membrane in internal reflection mode. Spectra were recorded on a Thermo Nicolet Nexus 670 spectrometer. The electrolyte was degassed by bubbling N$_2$ for 20 min before the measurement.

## Data availability

The data that support the plots within this paper are available from the corresponding author upon request. The source data underlying Figs. 2–5 are provided as a Source Data file. Source data and the optimized DFT computational models are provided with this paper.

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

## Acknowledgements

The authors are grateful to the National Natural Science Foundation of China (22271213 to B.Z. and 22001192 to C.B.). This work is also supported by the Fundamental Research Funds for the Central Universities of China, the National Supercomputing Center in Guangzhou (Sun Yat-Sen University) and the 1W1B beamline of the Beijing Synchrotron Radiation Facility.

## Author contributions

B.Z. and C.L. conceived the idea and directed the research. M.H., C.L., and B.Z. designed the experiments. M.H. and R.L. synthesized the materials and carried out the electrochemical experiments. M.H., R.L., and C.L. analysed the NMR data. C.C. contributed to the density functional theory calculations. C.L. wrote the paper. B.Z. revised the paper with comments from all the authors.

## Competing interests
