## [Peer Review File · Nature Communications]

Microenvironment regulation breaks the Faradaic efficiency–current density trade-off for electrocatalytic deuteration using D₂OREVIEWER COMMENTS

Reviewer #1 (Remarks to the Author):

Renewable electricity-powered deuteration using D₂O provides a mild and appealing D incorporation strategy. This demonstrated a combined strategy of controlling the nanotip-enhanced electric field and surfactant-modified interface microenvironment to improve the activity and selectivity of electrocatalytic deuteration reactions, which represents a novel and sustainable approach to delivering high FEs of EDRs at large current densities. Various spectroscopic techniques (XAS, Raman, FTIR, ICP-AES, EPR, etc.) and computational studies (DFT, AIMD, FEA) were conducted to substantiate their conclusions. Therefore, this manuscript is highly suitable for publication in Nature Communications.

The following comments should be addressed.

- In Fig. 2f, the author identifies different angles, such as 0, 77, 96, and 126.9, but the corresponding crystal planes and models for each angle are not clear.
- In the initial CV cycle (Fig. 3c), is the electrolyte the same? Why does the redox potential of the Cu species shift positively when 1a substrate is added?
- FE is not 100% in almost all cases in this article, what are the remaining byproducts? Is it all D₂?
- The diameter of Cu NTs should be labeled to ensure the enhanced electric field effect because the size of the Cu tips in the simulations is 50 ~ 5 nm.
- Check the errors in the article, for example, "AustedoTM" should be "AustedoTM".
- Some of the molecular migration pathways shown in Supplementary Fig. 3 need to be examined.
- On a similar note, the authors should mention what material was used as a counter electrode and membrane for the general procedure.

Reviewer #2 (Remarks to the Author):

The synthesis of deuterium compounds via electrocatalysis is of great importance. However, the D₂ evolution reaction is a severely competitive reaction that reduces the yield and Faraday efficiency of the target product. Zhang and Liu report a method that combines a nanotip catalyst with an interfacial microenvironment for the electrocatalytic deuteration of acrylonitrile in D₂O. The system exhibits a high FE at -100 mA/cm². The scalability for other deuteration reactions (including C-I, C=C, C-NO₂, and N-heterocycles) via this method is impressive. I propose to publish it in Nature Communications after solving the following problems.

1. Why only BTAB can enhance the FE of the reaction, while other surfactant have little effect?
2. The authors should add experimental evidence to confirm the advantage of coordination-unsaturated.
3. The authors should provide experimental details in the supporting information to indicate whether the FE was for the full process or for a period of time of the reaction.
4. Cu is a very easily oxidized material, so why are there no peaks of high-valent Cu measured in XPS?
5. The spectrum (Supplementary Fig. 16) seem to be further processed from Figs. 4a and 4b, and the authors may consider deleting this result, as the hydrophobic effect of the surfactant can be seen clearly through Figs. 4a and 4b.
6. Abundant coordination-unsaturated sites were found on high-curvature Cu Tips. Did the unsaturated Cu sites promote the electrocatalytic deuteration of acrylonitrile?

Reviewer #3 (Remarks to the Author):

He et al. report efficient electrochemical deuteration using a combination strategy of catalyst morphology engineering and surfactants to achieve high Faradaic efficiencies at high current densities, a great challenge in deuteration synthesis. The authors designed a free-standing copper nanotip electrode with quaternary ammonium-based surfactants to deuterate different acrylonitriles in D₂O with up to 80% Faradaic efficiency at 100 mA/cm². These results are solid, and the conclusions of the paper are well supported by the data.

Given the importance and novelty, I recommend this work to be published in Nature Communications after minor revisions according to the comments below.

1. When using the surfactant modification, the distance between the H₂O molecule and the electrode is significantly enlarged. Does it affect the deuteration process for organic substrates with D₂O as the D source?
2. In Figure 2a, the author calculated the number of hydrogen bonds that changed over time for two models, but the statistical ranges of hydrogen bond lengths and angles need to be clarified.
3. The value of the y-axis in Figure 2c is confused.
4. Please elaborate on why used 1,4-dioxane (diox)/0.5 M D₂O solution of K₂CO₃ as the electrolyte in this manuscript, instead of 0.5 M D₂O solution of K₂CO₃.
5. What is the peak at 2200 cm⁻¹ in the Raman in Figures 4a and 4b?
6. The error bars should be included in Figure 4e.
7. Descriptions for some experimental details are necessary. I suggest the author add the details about how the yields and FE of other deuterated reactions were determined in the Supporting Information.
8. Please check for errors in the font in the article, '1a' should be uniformly bolded.

A point-by-point response to the reviewers' comments

To reviewer 1:

Reviewer letter: Renewable electricity-powered deuteration using D₂O provides a mild and appealing D incorporation strategy. This demonstrated a combined strategy of controlling the nanotip-enhanced electric field and surfactant-modified interface microenvironment to improve the activity and selectivity of electrocatalytic deuteration reactions, which represents a novel and sustainable approach to delivering high FEs of EDRs at large current densities. Various spectroscopic techniques (XAS, Raman, FTIR, ICP-AES, EPR, etc.) and computational studies (DFT, AIMD, FEA) were conducted to substantiate their conclusions. Therefore, this manuscript is highly suitable for publication in Nature Communications.

Answer: We highly appreciate the reviewer's very positive comments on our work. Regarding the other comments or suggestions of the reviewer, we have provided point-by-point responses. To save the reviewer's valuable time, key revisions are displayed on a yellow background in the revised manuscript and Supplementary Information. We are sure that the quality of this work will be greatly improved after being revised.

Comment 1: In Figure 2f, the author identifies different angles, such as 0, 77, 96, and 126.9, but the corresponding crystal planes and models for each angle are not clear.

Answer: Thank you for the comments. By measuring the dihedral angle of the relaxed model, we defined the angles of (111)/(111) as 77 degrees, (100)/(100) as 96 degrees, and (111)/(100) as 126.9 degrees. The corresponding revisions have been included in the revised Supplementary Information (Supplementary Note 3).

Comment 2: In the initial CV cycle (Figure 3c), is the electrolyte the same? Why does the redox potential of the Cu species shift positively when **1a** substrate is added?

Answer: Thank you for your comment. The CV test was carried out in a mixed solution of dioxane and 0.5 M K₂CO₃ in D₂O (2:5 v/v, 7 mL), which was the same as that used for the deuteration reaction.

The redox peak of the metal electrode from the CV curves shows the reduction and oxidation processes via the loss and gain of electrons. The adsorption of organic molecules may cause a change in the electronic structure of the catalyst (*Nat. Commun.* 2018, 9, 1610; *J. Am. Chem. Soc.* 2023, 145, 6622), thus influencing the redox potentials of the reduction and oxidation of metals. In our work, the redox potential shift of Cu species after the addition of **1a** may be due to electronic interactions between the Cu electrode and the adsorbed **1a**. Because of the strong electron-withdrawing property of the C≡N group, a positive shift in the deduction potential of Cu is observed. In addition, the obvious **1a** reduction

peak in the CV curve (Figure R1a) (Figure 3c in the revised manuscript) and the visible redshift of the $\text{C}\equiv\text{N}$ vibration ($\nu_{\text{C}\equiv\text{N}}$) of **1a** over the Cu electrode compared with that of pure **1a** in the time-dependent in situ Raman spectra both indicate that **1a** is adsorbed on the Cu surface under bias potentials (Figure R1b) (Supplementary Figure 26 in the revised Supplementary Information), which may cause a shift in the redox potentials of the oxidation and reduction processes of Cu.

Figure R1 (a) CV curve of Cu NTs at a scan rate of 3 mV s^{-1} in a mixed solution of diox/ $0.5 \text{ M K}_2\text{CO}_3$ (2:5 v/v, 7 mL) with **1a**. (b) In situ Raman spectra of a mixed solution of diox/ $0.5 \text{ M K}_2\text{CO}_3$ for the electrocatalytic deuteration of **1a** over Cu NTs at $-1.3 \text{ V vs. Hg/HgO}$.

Comment 3: FE is not 100% in almost all cases in this article, what are the remaining byproducts? Is it all D_2 ?

Answer: Thank you very much for the comments. We used gas chromatography–mass spectrometry (GC–MS) to qualitatively analyse all the possible products of the deuteration of **1a**. As shown in Figure R2, only the desired product **2a** was detected without any other byproducts, such as the corresponding amide or carboxylic acid, as mentioned in the literature (*Nat. Commun.* 2022, 13, 5951). Additionally, amide and carboxylic acid byproducts were not detected by liquid chromatography (LC). Therefore, we deduce that D_2 is the only remaining byproduct of the deuteration reaction, which decreases the FE of **2a**.

Figure R2 Detection of the reaction system for the deuteration of **1a** by using GC–MS.

Comment 4: The diameter of Cu NTs should be labeled to ensure the enhanced electric field effect because the size of the Cu tips in the simulations is $50 \sim 5$ nm.

Answer: Thank you for the kind comments. The TEM image in Figure R3 suggests a tip diameter in the range of ~ 14 nm, indicating that the radius of curvature of the Cu NTs is ~ 7 nm, which is much smaller than that of the Cu NRs and Cu NSs (Supplementary Figure 12 in the revised Supplementary Information). Thus, compared with Cu NRs and NSs, Cu NTs have markedly enhanced electric fields. We have added the diameter of the Cu NTs to the revised Supplementary Information (Supplementary Figure 6b).

Figure R3 TEM image of Cu NTs with a radius of curvature of ~ 7 nm.

Comment 5: Check the errors in the article, for example, “AustedoTM” should be “AustedoTM”.

Answer: Thank you very much for this valuable suggestion. We have changed “AustedoTM” to “AustedoTM” in the revised manuscript. We have also carefully checked the typos and grammar throughout the manuscript, and the revisions are displayed with a yellow background in the revised manuscript.

Comment 6: Some of the molecular migration pathways shown in Supplementary Fig. 3 need to be examined.

Answer: We acknowledge the reviewer’s comment. We have revised the molecular migration pathway in the solution without surfactants in Supplementary Figure 3b. The resulting Supplementary Figure 3b was extracted and is shown in Figure R4.

Figure R4 The transport of **1a** in solution without BTAB.

Comment 7: On a similar note, the authors should mention what material was used as a counter electrode and membrane for the general procedure.

Answer: Thank you for the kind comment. A carbon rod was used as the counter electrode, and Nafion 117 (NF117) was used as the membrane. We have added the related experimental details to the revised manuscript. The corresponding revisions were extracted and are shown as follows:

“The as-prepared electrocatalyst, carbon rod, and Hg/HgO were used as the working electrode, counter electrode, and reference electrode, respectively. The cathodic and anodic chambers were separated by a Nafion 117 (NF117) membrane.”

We highly appreciate the reviewer’s thorough reading and constructive comments/suggestions about our manuscript.

To reviewer 2:

Reviewer letter: The synthesis of deuterium compounds via electrocatalysis is of great importance. However, the D₂ evolution reaction is a severely competitive reaction that reduces the yield and Faraday efficiency of the target product. Zhang and Liu report a method that combines a nanotip catalyst with an interfacial microenvironment for the electrocatalytic deuteration of arylacetonitrile in D₂O. The system exhibits a high FE at -100 mA⁻². The scalability for other deuteration reactions (including C–I, C=C, C–NO₂, and N-heterocycles) via this method is impressive. I propose to publish it in Nature Communications after solving the following problems.

Answer: We highly appreciate the reviewer's very positive comments on our manuscript. Regarding the other comments or suggestions of the reviewer, we have provided point-by-point responses. To save the reviewer's valuable time, key revisions are displayed on a yellow background in the revised manuscript and Supplementary Information. We are sure that the quality of this work will be greatly improved after being revised.

Comment 1: Why only BTAB can enhance the FE of the reaction, while other surfactant have little effect?

Answer: Thank you for this comment. The introduction of BTAB with a medium-length tail can increase the FE up to 80% and requires the lowest cell voltage at -100 mA cm⁻². However, the use of tetramethyl ammonium bromide (TMAB) and octadecyl trimethyl ammonium bromide (OTAB) had no obvious or even negative effect on the deuteration of **1a**. We speculate that the smallest TMAB may block more active sites on Cu and be ineffective at breaking hydrogen bond networks, whereas the longer alkyl chain of OTAB may affect the migration of **1a**, thus reducing the reaction efficiency. This effect has also been reported in the literature on the electrocatalytic semihydrogenation of alkynes by Li and coworkers (*J. Am. Chem. Soc.* 2023, 145, 6516–6525.).

Comment 2: Abundant coordination-unsaturated sites were found on high-curvature Cu Tips. Did the unsaturated Cu sites promote the electrocatalytic deuteration of arylacetonitrile? The authors should add experimental evidence to confirm the advantage of coordination-unsaturated.

Answer: We acknowledge the reviewer's kind comments. The answer is that unsaturated Cu sites can promote the electrocatalytic deuteration of arylacetonitrile. We will explain this below. Enhancing the adsorption of substrates on the electrode surface is usually favourable for accelerating their activation, thus boosting electrocatalytic deuteration reactions (*Nat. Commun.* 2022, 13, 5951.). In addition, introducing low-coordination sites into electrode materials is an efficient way to greatly increase the intrinsic activity of active sites by optimizing adsorption (*Nat. Energy* 2019, 4, 690–699.). To validate the role of the coordination-unsaturated sites of Cu NTs in promoting the electrocatalytic deuteration of arylacetonitrile, we synthesized Cu-H₂ NTs with fewer coordination-unsaturated sites than Cu NTs (Figure

R5a, see Figure 3b in the revised manuscript) for comparison with the electrocatalytic deuteration of **1a**. Potential screening experiments showed that both the yield and FE of **2a** were greater for Cu NTs than for Cu-H₂ NTs at the same potentials (Figure R5b and R5c, see Supplementary Figure 9 in the revised Supplementary Information). The yield of **2a** was also higher for the Cu NTs than for the Cu-H₂ NTs at -100 mA cm⁻² (Figure R5c). These results demonstrate the promoting effect of more low-coordination sites on the electrocatalytic deuteration of **1a**.

Figure R5 (a) EXAFS spectra of Cu NTs and Cu-H₂ NTs. Potential-dependent yield and FE of **2a** over (b) Cu NTs and (c) Cu-H₂ NTs. (d) Yield of **2a** at -100 mA cm⁻² over Cu NTs and Cu-H₂ NTs.

Comment 3: The authors should provide experimental details in the supporting information to indicate whether the FE was for the full process or for a period of time of the reaction.

Answer: Thank you for the comments. The FE for the full process of deuteration was calculated by dividing the actually formed moles of the deuterated product by the theoretically formed moles of the deuterated product under the theoretical Coulomb charge for full conversion of reactants. We have added details related to FEs to the revised manuscript. The corresponding revisions were extracted and are shown as follows:

*“The yield and FE, yield rate and TOF (**2a** as examples) were for the whole process of deuteration, which are determined using dodecane as an internal standard and calculated using equations (1) - (4) below.”*

Comment 4. Cu is a very easily oxidized material, so why are there no peaks of high-valent Cu measured in XPS?

Answer: We highly appreciate the reviewer's comments. X-ray photoelectron spectroscopy (XPS) is a useful analytical technique for characterizing elements and their chemical states on the surface of materials. Since catalysts are exposed to the atmosphere before XPS analysis, the surface is prone to oxidation. To obtain true information about the sample surface, argon ion etching is commonly used in XPS test (*J. Am. Chem. Soc.* 2003, 125, 5998.). Therefore, we used argon ion etching 3 times before XPS testing to obtain the real chemical valence state of the Cu NTs. The relevant details have been added to the revised Supplementary Information (Supplementary Note 5). The corresponding revisions were extracted and are shown as follows:

"To obtain the real chemical valence state of the Cu NTs, the sample was treated through argon ion etching 3 times (20 nm) before being subjected to XPS test."

Comment 5: The spectrum (Supplementary Fig. 16) seem to be further processed from Figs. 4a and 4b, and the authors may consider deleting this result, as the hydrophobic effect of the surfactant can be seen clearly through Figs. 4a and 4b.

Answer: We acknowledge the reviewer's kind suggestion. We have deleted the spectrum of Supplementary Figure 16 in the revised Supplementary Information.

We highly appreciate the reviewer's thorough reading and constructive comments/suggestions about our manuscript.

To reviewer 3:

Reviewer letter: He et al. report efficient electrochemical deuteration using a combination strategy of catalyst morphology engineering and surfactants to achieve high Faradaic efficiencies at high current densities, a great challenge in deuteration synthesis. The authors designed a free-standing copper nanotip electrode with quaternary ammonium-based surfactants to deuterate different aryl acetonitriles in D₂O with up to 80% Faradaic efficiency at 100 mA cm⁻². These results are solid, and the conclusions of the paper are well supported by the data. Given the importance and novelty, I recommend this work to be published in Nature Communications after minor revisions according to the comments below.

Answer: We highly appreciate the reviewer's very positive comments on our manuscript. Regarding the other comments or suggestions of the reviewer, we have provided point-by-point responses. To save the reviewer's valuable time, key revisions are displayed on a yellow background in the revised manuscript and Supplementary Information. We are sure that the quality of this work will be greatly improved after being revised.

Comment 1: When using the surfactant modification, the distance between the H₂O molecule and the electrode is significantly enlarged. Does it affect the deuteration process for organic substrates with D₂O as the D source?

Answer: We acknowledge the reviewer's kind comments. For electrochemical deuteration reactions using D₂O as the D source, the concentration of D₂O around the electrode is far greater than that of the reactant, which often leads to a serious D₂ evolution reaction (DER) at high current densities. In our work, the addition of surfactants can create a hydrophobic interface microenvironment and lower the amount of D₂O near the electrode surface, thereby suppressing the DER. As shown in Figures R6a and 6b (Figure 4c in the revised manuscript and Supplementary Figure 21 in the revised Supporting Information), the Tafel slope for the DER increases from 224 to 295 mV dec⁻¹, and a more negative potential is needed to achieve a benchmark current density of -10 mA cm⁻² after introducing the optimal amount of BTAB, implying a suppressed DER. However, the amount of D₂O near the surface of the electrode is still sufficient for the deuteration of **1a** under these circumstances; therefore, the FE and yield rate of **2a** are improved (Figure R6c) (Figure 3d in the revised manuscript). These results demonstrate that the addition of an appropriate amount of BTAB can promote the deuteration reaction and inhibit the competitive DER, which is beneficial for the deuteration reaction.

Figure R6 (a) Tafel slopes of Cu NTs in 0.5 M K_2CO_3 (D_2O) with and without 1.0 mM BTAB. (b) LSV curves of Cu NTs in 0.5 M K_2CO_3 (D_2O) with and without BTAB. (c) Comparison of FEs and yield rates of **2a** over Cu NTs at -100 mA cm^{-2} with and without BTAB.

Comment 2: In Figure 2a, the author calculated the number of hydrogen bonds that changed over time for two models, but the statistical ranges of hydrogen bond lengths and angles need to be clarified.

Answer: We acknowledge the reviewer's kind suggestion. In the process of hydrogen bond calculation of the AIMD model, the bond length between H and O and C in the model ranges from 1.2 \AA to 4 \AA , and the bond angle cut-off is 40 degrees. The details have been added to the Supplementary Information (Supplementary Note 1). The corresponding revisions were extracted and are shown as follows:

"In the process of hydrogen bond calculation of the AIMD model, the bond length between H and O, and C elements in the model ranges is from 1.2 \AA to 4 \AA , and the bond angle cut-off is 40 degrees."

Comment 3: The value of the y-axis in Figure 2c is confused.

Answer: Thank you for the comment. We have modified the y-axis of Figure 2c in the revised manuscript (Figure R7).

Figure R7 Mass density of H_2O molecules versus distance above the electrode surface with and without BTAB.

Comment 4: Please elaborate on why used 1,4-dioxane (diox)/0.5 M D₂O solution of K₂CO₃ as the electrolyte in this manuscript, instead of 0.5 M D₂O solution of K₂CO₃.

Answer: Thank you for the comment. The addition of an appropriate amount of 1,4-dioxane (diox) to the electrolyte improves the solubility of organic substrates, which is beneficial for deuteration reactions. We have added these descriptions to the revised manuscript. The corresponding revisions were extracted and are shown as follows:

The addition of the right amount of 1,4-dioxane to the electrolyte improved the solubility of the organic substrates, which is beneficial for the deuteration reactions.

Comment 5: What is the peak at 2200 cm⁻¹ in the Raman in Figures 4a and 4b?

Answer: Thank you for the comment. The peak is assigned to the CO₂ stretching vibrational modes (*Angew. Chem. Int. Ed.* 2023, 62, e202309875.). We have labelled the peaks in Figures 4a and 4b in the revised manuscript.

Comment 6: The error bars should be included in Figure 4e.

Answer: We deeply acknowledge the reviewer's kind suggestion. We have added error bars to Figure 4e in the revised manuscript, which can also be seen below (Figure R8).

Figure R8 Field-induced BTA⁺ concentration loss in the electrolytes caused by the use of Cu NTs, Cu NRs, and Cu NSs.

Comment 7: Descriptions for some experimental details are necessary. I suggest the author add the details about how the yields and FE of other deuterated reactions were determined in the Supporting Information.

Answer: Thank you for the kind suggestions. Gas chromatography (GC) is a powerful analytical technique used to separate, identify, and quantify individual chemical components in complex mixtures. According to GC chromatograms, the identification of products can be confirmed by comparison of their GC retention times depending on the boiling point of the different compounds, and the conversion yield and

product selectivity can be determined by comparing their peak areas. Commonly, a calibration curve of standard compounds with known concentrations is used to determine the concentration of a sample (*Nat. Catal.* 2022, 5, 66–73; *ACS Catal.* 2018, 8, 8396–8405.). Because the boiling points of the other deuterated products are within the range measured by GC, we used GC to quantify the moles of the obtained deuterated products according to their standard calibration curves. Then, we can calculate the yields and FEs of the other deuterated reactions according to equations (1) - (2). The standard calibration curves are shown in Figure R9 and have also been added to the revised Supplementary Information (Supplementary Figure 31).

$$\text{Yield (\%)} = \frac{n \text{ (obtained products)}}{n \text{ (theoretically formed products)}} \times 100\% \quad (1)$$

$$\text{FE (\%)} = \frac{4 \times n \times 96480}{Q \text{ (theoretically values)}} \times 100\% \quad (2)$$

Figure R9 Standard calibration curves for the quantitative analysis of other reactions with dodecane as an internal standard.

Comment 8: Please check for errors in the font in the article, '1a' should be uniformly bolded.

Answer: Thank you for the kind suggestion. We have used the bolded font of **1a** in the revised manuscript. We have also carefully checked the typos and grammar throughout the manuscript, and the revisions are displayed on a yellow background.

We highly appreciate the reviewer's thorough reading and constructive comments/suggestions about our manuscript.

REVIEWERS' COMMENTS

Reviewer #1 (Remarks to the Author):

The manuscript has been well revised for final publication.

Reviewer #2 (Remarks to the Author):

This manuscript could be accepted without any further revisions.

Reviewer #3 (Remarks to the Author):

My comments have been well addressed. I recommend its publication without further revisions.

A point-by-point response to the reviewers

To reviewer 1:

Reviewer letter: The manuscript has been well revised for final publication.

Answer: We highly appreciate the reviewer's very positive comments on our work. We are sure that the quality of this work has been greatly improved according to these comments and suggestions.

To reviewer 2:

Reviewer letter: This manuscript could be accepted without any further revisions.

Answer: We highly appreciate the reviewer for his/her positive comments on our revised manuscript. We are sure that the quality of this work has been greatly improved according to these comments.

To reviewer 3:

Reviewer letter: My comments have been well addressed. I recommend its publication without further revisions.

Answer: We highly appreciate the reviewer's positive comments on our revised manuscript. We are sure that the quality of this work has been greatly improved according to these comments.